# A Proposed Device for Controlling the Flow of Information Based on Weyl Fermions

**DOI:** 10.3390/s24113361

**Published:** 2024-05-24

**Authors:** Georgios N. Tsigaridas, Aristides I. Kechriniotis, Christos A. Tsonos, Konstantinos K. Delibasis

**Affiliations:** 1Department of Physics, School of Applied Mathematical and Physical Sciences, National Technical University of Athens, GR-15772 Athens, Greece; 2Department of Physics, University of Thessaly, GR-35100 Lamia, Greece; akechriniotis@uth.gr (A.I.K.); christostsonos@uth.gr (C.A.T.); 3Department of Computer Science and Biomedical Informatics, University of Thessaly, GR-35131 Lamia, Greece; kdelimpasis@dib.uth.gr

**Keywords:** parallel switch, Weyl fermions, Weyltronics, electromagnetic fields, electric field sensors, manipulation of Weyl particles, electromagnetic interactions of high energy particles

## Abstract

In this work we propose a novel device for controlling the flow of information using Weyl fermions. Based on a previous work by our group, we show that it is possible to fully control the flow of Weyl fermions on several different channels by applying an electric field perpendicular to the direction of motion of the particles on each channel. In this way, we can transmit information as logical bits, depending on the existence or not of a Weyl current on each channel. We also show that the response time of this device is exceptionally low, less than 1 ps, for typical values of its parameters, allowing for the control of the flow of information at extremely high rates of the order of 100 Petabits per second. Alternatively, this device could also operate as an electric field sensor. In addition, we demonstrate that Weyl fermions can be efficiently guided through the proposed device using appropriate magnetic fields. Finally, we discuss some particularly interesting remarks regarding the electromagnetic interactions of high-energy particles.

## 1. Introduction

In 1929, German physicist and mathematician Hermann Weyl predicted the existence of massless fermions that carry an electric charge, named as Weyl fermions [1]. The nature of these particles suggests that they possess a high degree of mobility, moving very quickly on the surface of a crystal with no backscattering, offering substantially higher efficiency and lower heat generation compared to conventional electronics. Furthermore, these particles possess a special form of chirality with their spin being either parallel or anti-parallel to their direction of motion, referred to as positive and negative helicity, respectively. At the time of writing no such particles have been observed in nature, as free particles. 

However, in 2015, an international research team led by scientists at Princeton University detected Weyl fermions as emergent quasiparticles in synthetic crystals of the semimetal tantalum arsenide (TaAs) [2]. Independently, in the same year a research team led by M. Soljacic at the Massachusetts Institute of Technology, also observed Weyl-like excitations in photonic crystals [3]. These discoveries have offerred the opportunity to design and develop novel devices based on Weyl fermions instead of electrons, leading to a new branch of electronics, Weyltronics [4,5,6,7,8,9,10]. It should also be noted that, recently, thin films of Weyl semimetals have also been realized [4], facilitating further the development of Weyltronic devices in photonics [11,12,13,14,15,16,17], spintronics [18,19,20,21,22], and other applications [23,24,25,26].

In this work we describe the principles of operation of a novel device for controlling the flow of information using Weyl fermions, referred to as the Weyl Parallel Switch or WPS. This device is expected to offer significant advantages over similar devices based on conventional electronics, such as exceptionally low response time, increased power efficiency and extremely high bandwidth. Furthermore, due to the remarkable property of Weyl particles to be able to exist in the same quantum state in a wide variety of electromagnetic fields [27,28], we anticipate that the proposed device will offer enhanced robustness against electromagnetic perturbations, enabling it to be used efficiently even in environments with high levels of electromagnetic noise. Therefore, the WPS is expected to play an important role in the emerging field of Weyltronics [4,5,6,7,8,9,10] and find significant applications in several fields, such as telecommunications, signal processing, classical and quantum computing, etc. In addition, in Section 3, we calculate the magnetic fields that could be used to fully control the transverse spatial distribution of Weyl fermions, which is expected to be particularly useful regarding the practical applications of Weyl particles. Finally, in the Appendix A we discuss a particularly interesting remark regarding the electromagnetic interactions of high-energy particles.

## 2. Design and Characteristics of the Proposed Device

For designing the proposed device, we rely on the theory developed in a previous work by our group [27] where we have shown that Weyl particles can exist in localized states, even in the absence of electromagnetic fields. Furthermore, the localization of Weyl fermions can be easily adjusted using simple electric fields perpendicular to the direction of motion of the particles. This is clearly shown in Figures 4 and 5 in [27]. It should also be mentioned that we choose the electric field to be perpendicular to the direction of motion of the particles because an electric field parallel to the direction of motion of Weyl fermions would only affect their energy and not their localization, as is also discussed in [27].

In more detail, we show that the radius of the region where the Weyl particle is confined is given by the formula (Equation (36) in [27])
(1)$$r\left(t\right)=\frac{r_{0}}{1\pm 2qr_{0}\left|E\right|t}$$
where $r_{0}$ is the initial value of the radius, prior to the application of the electric field, q is the electric charge of the particle and $\left|E\right|$ is the magnitude of the electric field. The sign in the denominator of Equation (1) depends on the direction of the electric field relative to the angular velocity of the particles. In more detail, in the case of Weyl particles with positive helicity, the radius decreases if the electric field is anti-parallel to the vector of the angular velocity of the particles; otherwise, the radius increases. The opposite is true for Weyl particles with negative helicity. 

According to Equation (1) if the radius increases with time, it becomes infinite after a time interval equal to (Equation (37) in [27])
(2)$$\Delta t=\frac{1}{2qr_{0}\left|E\right|}$$

If the electric field continues to be applied, the radius becomes negative and decreases in magnitude, implying that the Weyl particle becomes localized again with the vector of the angular velocity pointing to the opposite direction. 

It should also be noted that Equations (1) and (2) are expressed in natural units, where ℏ=c=1. In S.I. units they take the form:(3)$$r\left(t\right)=\frac{r_{0}}{1\pm \left(2qr_{0}\left|E\right|/\hbar \right)t}$$
and
(4)$$\Delta t=\frac{\hbar }{2qr_{0}\left|E\right|}$$
respectively.

From the above analysis it becomes evident that it is possible to fully control the propagation of Weyl particles using simple electric fields perpendicular to their direction of motion. In more detail, if we assume that a Weyl particle moves initially on a straight line and an electric field perpendicular to its direction of motion is applied at t=0, then the particle becomes localized and is confined to a region of radius $r_{0}$ after a time interval given by Equation (4).

This behavior can be utilized for developing a device for controlling the flow of information on multiple channels simultaneously. This device, henceforth called a Weyl Parallel Switch or WPS, is shown in the schematic diagram of Figure 1.

The proposed WPS consists of a slab of a material supporting Weyl particles. An array of capacitors is constructed on this material to control the motion of Weyl fermions on each channel by adjusting the voltage applied to the capacitor corresponding to this channel. The capacitors are placed in a way that the electric field is perpendicular to the direction of motion of the particles, as shown in Figure 1. If we assume that no voltage is applied to the capacitors, Weyl particles move along straight lines on each channel, transferring a current to the output of the channel. On the other hand, if a voltage is applied to the capacitor, the resulting electric field, which is perpendicular to the direction of motion of the particles, will confine them to a circular region of radius $r_{0}$. Consequently, no current will be delivered to the output of this channel. It should also be mentioned that we have introduced an insulating layer between the channels to avoid any interactions between Weyl particles propagating in different channels, although these interactions are expected to be exceptionally weak.

Therefore, it is possible to control the flow of the current on each channel through the voltage applied to the capacitor corresponding to this channel. Consequently, we can control the flow of information through the channels, supposing, for example, that the presence of a current corresponds to a logical “one” and its absence to a logical “zero”, as shown in Figure 1. It should also be noted that, using the proposed device, we can control the flow of many bits of information simultaneously, equal to the number of channels. 

The time required for the confinement of Weyl particles can be easily calculated using Equation (4), which can also be written in the following form:(5)$$\frac{qr_{0}\Delta V\Delta t}{d}=\frac{\hbar }{2}$$
where ΔV is the voltage applied to each capacitor and d is the distance between the plates of the capacitor. 

As far as the width of the channel is concerned, it can be determined by the radius of the area where Weyl particles are confined, which, according to Equation (5), is given by the formula:(6)$$r_{0}=\frac{\hbar }{2q\left|E\right|\Delta t}$$

Obviously, the width of the channel is equal to $2r_{0}$. Assuming that the charge of the particles is equal to the electron charge, it is easy to estimate the numerical value of the channel width as a function of the amplitude of the electric field $\left|E\right|$ and its application time Δt, given by the following formula:(7)$$w_{ch}=2r_{0}=\frac{\hbar }{q\left|E\right|\Delta t}=\frac{6.58}{\left|E\right|\Delta t}\times 10^{-16}m=\frac{0.658}{\left|E\right|\Delta t}fm$$
which is exceptionally small. For example, if $\left|E\right|=10^{4}\;V/m$ and Δt=1 ps, the above formula implies that $w_{ch}=65.8\;\mathrm{nm}$. Obviously, the channel width can become much smaller if the amplitude or the application time of the electric field is increased. 

As far as the type of material which is preferable for the proposed device is concerned, we would like to mention that we have made no assumptions regarding the properties of the material. Consequently, any material supporting Weyl particles should be suitable for the proposed device. For practical reasons, we would prefer materials that can be shaped in the form of thin films [4], as it is also mentioned in the introduction. 

Furthermore, assuming that the charge of Weyl particles is equal to the electron charge, the distance between the plates of the capacitor is d=1 mm and the voltage applied to each capacitor is ΔV=10 V, Equation (5) implies that the time interval required to confine Weyl particles to a region of radius $r_{0}=50\;\mathrm{nm}$ is equal to Δt=0.658 ps. Consequently, the response time of the proposed device is exceptionally low for typical values of its parameters, increasing its efficiency further. Here, it should also be mentioned that, if a voltage with opposite polarity is applied to a channel with confined particles and no current flow, then Weyl particles in this channel will become again delocalized and the current will reappear at the output of this channel. Obviously, the voltage must have the same magnitude and be applied for the same amount of time with the one used for confining Weyl particles.

Thus, it is possible to switch between logical “zeros” and “ones”—and vice versa—with a response time of the order of 1 ps for typical values of the parameters. In addition, assuming that the width of each channel is of the order of $2r_{0}$ and the full width of the material used in this device is of the order of 1 cm, we understand that the device can support up to $10^{5}$ channels. This practically means that, using the WPS, we can control the flow of information at a rate of the order of $10^{17}$ bits per second, 100 Pbps, which is exceptionally difficult to achieve using conventional electronics.

Furthermore, the use of Weyl particles instead of electrons for transporting information offers higher transfer speeds, twice as fast as in graphene and up to 1000 times higher compared to conventional semiconductors [2,10], and more efficient energy flow, substantially reducing heat generation due to collisions with the ions of the lattice. This practically means that the energy consumption of the WPS, and any other device based on Weyl particles, is expected to be orders of magnitude lower than the consumption of devices based on conventional electronics. 

In addition, as shown in [27,28], Weyl particles have the remarkable property to be able to exist in the same quantum state under a wide variety of electromagnetic fields. Specifically, as shown in [27], the quantum state of Weyl particles will not be affected by the presence of a wide variety of electromagnetic fields, which, in S.I. units, are given by the following formulae:(8)$$\begin{array}{c} E_{s}\left(r,t\right)=-\frac{1}{q}\left[\sin\theta \cos\varphi \frac{1}{c}\frac{\partial s}{\partial t}+\frac{\partial s}{\partial x}+\frac{s}{c}\left(\cos\theta \cos\varphi \frac{d\theta }{dt}-\sin\theta \sin\varphi \frac{d\varphi }{dt}\right)\right]i\\ \;\;\;\;\;\;\;\;\;\;\;\;\;-\frac{1}{q}\left[\sin\theta \sin\varphi \frac{1}{c}\frac{\partial s}{\partial t}+\frac{\partial s}{\partial y}+\frac{s}{c}\left(\cos\theta \sin\varphi \frac{d\theta }{dt}+\sin\theta \cos\varphi \frac{d\varphi }{dt}\right)\right]j\\ \;\;\;\;\;\;\;\;\;\;\;\;\;-\frac{1}{q}\left(\cos\theta \frac{1}{c}\frac{\partial s}{\partial t}+\frac{\partial s}{\partial z}+\frac{s}{c}\sin\theta \frac{d\theta }{dt}\right)k\\ B_{s}\left(r,t\right)=\frac{1}{qc}\left(-\sin\theta \sin\varphi \frac{\partial s}{\partial z}+\cos\theta \frac{\partial s}{\partial y}\right)i+\frac{1}{qc}\left(\sin\theta \cos\varphi \frac{\partial s}{\partial z}-\cos\theta \frac{\partial s}{\partial x}\right)j\\ \;\;\;\;\;\;\;\;\;\;\;\;\;+\frac{1}{qc}\sin\theta \left(-\cos\varphi \frac{\partial s}{\partial y}+\sin\varphi \frac{\partial s}{\partial x}\right)k \end{array}$$
where θ, φ are the polar and azimuthal angle, respectively, corresponding to the propagation direction of the particles, and s is an arbitrary real function of the spatial coordinates and time. It should also be mentioned that, if the above electromagnetic fields are given in S.I. units, the function $s\left(r,t\right)$ should be expressed in joules. As an example, we suppose that Weyl particles move at the plane θ=π/2. Then, the electromagnetic fields given by Equation (8) take the simplified form:(9)$$\begin{array}{c} E_{s}\left(r,t\right)=-\frac{1}{q}\left(\cos\varphi \frac{1}{c}\frac{\partial s}{\partial t}+\frac{\partial s}{\partial x}-\frac{s}{c}\sin\varphi \frac{d\varphi }{dt}\right)i-\frac{1}{q}\left(\sin\varphi \frac{1}{c}\frac{\partial s}{\partial t}+\frac{\partial s}{\partial y}+\frac{s}{c}\cos\varphi \frac{d\varphi }{dt}\right)j\\ B_{s}\left(r,t\right)=-\frac{1}{qc}\sin\varphi \frac{\partial s}{\partial z}i+\frac{1}{qc}\cos\varphi \frac{\partial s}{\partial z}j+\frac{1}{qc}\left(-\cos\varphi \frac{\partial s}{\partial y}+\sin\varphi \frac{\partial s}{\partial x}\right)k \end{array}$$
where the azimuthal angle φ is constant in the case of free particles when no voltage is applied. However, in the case of particles confined by a voltage ΔV, its evolution is governed by the following differential equation:(10)$$\frac{d^{2}\varphi }{dt^{2}}=-2q\frac{c}{\hbar }\frac{\Delta V}{d}$$
leading to Equation (5), describing the time dependence of the radius of the confined particle as a function of the applied voltage. Thus, the WPS is expected to offer enhanced robustness against electromagnetic perturbations caused by the aforementioned electromagnetic fields, since the quantum state of Weyl particles will not be affected by the presence of the wide variety of electromagnetic fields given by the above formulae. 

Finally, it should be mentioned that the proposed device could also operate as an electric field sensor. In more detail, the presence of an electric field, perpendicular to the Weyl current propagating in a specific channel of the device, could alter the propagation direction of the Weyl particles, interrupting the current in this channel. Specifically, Equation (7) implies that, for a channel width $w_{ch}$ equal to 658 nm, the WPS could detect an electric field of magnitude $\left|E\right|=10^{-6}\;V/m$ within a time interval of 1 ms. Obviously, the sensitivity of the device as an electric field sensor will improve, increasing the width of the channel. 

## 3. Controlling the Spatial Distribution of Weyl Particles Using Appropriate Magnetic Fields

It is easy to verify that the spinor:(11)$$\psi_{m}=f\left(x,y\right)\left(\begin{array}{c} 1\\ 0 \end{array}\right)\exp\left[iE_{0}\left(z-t\right)\right]$$
describing Weyl particles with positive helicity moving along the +z direction with energy $E_{0}$ and transverse spatial distribution given by the arbitrary real function $f\left(x,y\right)$, is solution to the Weyl equation in the form given by Equation (1) in [27] for the following 4-potential:(12)$$\left(a_{0},a_{1},a_{2},a_{3}\right)=\left(0,\frac{1}{f}\frac{\partial f}{\partial y},-\frac{1}{f}\frac{\partial f}{\partial x},0\right)$$

The electromagnetic field corresponding to the above 4-potential can be easily calculated through the formulae [29,30]:(13)$$E=-\nabla U-\frac{\partial A}{\partial t},\;B=\nabla \times A$$
where $U=a_{0}/q$ is the electric potential and $A=-\left(1/q\right)\left(a_{1}i+a_{2}j+a_{3}k\right)$ is the magnetic vector potential. Using Equation (13), we obtain the electromagnetic field corresponding to the above 4-potential:(14)$$E=0,B=-\frac{1}{q}\frac{1}{f^{2}}\left[{\left(\frac{\partial f}{\partial x}\right)}^{2}+{\left(\frac{\partial f}{\partial y}\right)}^{2}-f\left(\frac{\partial^{2}f}{\partial x^{2}}+\frac{\partial^{2}f}{\partial y^{2}}\right)\right]k$$

Thus, it is possible to fully control the transverse spatial distribution of Weyl particles using the magnetic field given by Equation (14) along their propagation direction. 

As an example, we consider that $f\left(x,y\right)$ is given by a generalized super-gaussian orthogonal distribution of the form:(15)$$f\left(x,y\right)=\exp\left[-{\left(\frac{{\left(x-x_{0}\right)}^{2}}{\sigma_{x}^{2}}\right)}^{p_{x}}-{\left(\frac{{\left(y-y_{0}\right)}^{2}}{\sigma_{y}^{2}}\right)}^{p_{y}}\right]$$
where $x_{0},y_{0}$ are arbitrary real constants corresponding to the center of the distribution and $\sigma_{x},\sigma_{y},p_{x},p_{y}$ are arbitrary positive constants corresponding to the widths and the exponents of the distribution, respectively. According to Equation (12), the 4-potential corresponding to the above distribution is the following:(16)$$\left(a_{0},a_{1},a_{2},a_{3}\right)=\left(0,-\frac{2p_{y}\left(y-y_{0}\right)}{\sigma_{y}^{2}}{\left(\frac{{\left(y-y_{0}\right)}^{2}}{\sigma_{y}^{2}}\right)}^{-1+p_{y}},\frac{2p_{x}\left(x-x_{0}\right)}{\sigma_{x}^{2}}{\left(\frac{{\left(x-x_{0}\right)}^{2}}{\sigma_{x}^{2}}\right)}^{-1+p_{x}},0\right)$$
and the magnetic field given by Equation (14) becomes:(17)$$B=-\frac{2}{q}\left[\frac{p_{x}\left(-1+2p_{x}\right)}{\sigma_{x}^{2}}{\left(\frac{{\left(x-x_{0}\right)}^{2}}{\sigma_{x}^{2}}\right)}^{-1+p_{x}}+\frac{p_{y}\left(-1+2p_{y}\right)}{\sigma_{y}^{2}}{\left(\frac{{\left(y-y_{0}\right)}^{2}}{\sigma_{y}^{2}}\right)}^{-1+p_{y}}\right]k$$

Similarly, in the case of Weyl fermions with negative helicity, described by Equation (2) in [27], the spinor corresponding to particles moving along the +z direction with energy $E_{0}$ and transverse spatial distribution given by the function $f\left(x,y\right)$ is the following:(18)$$\psi_{m}^{\prime }=f\left(x,y\right)\left(\begin{array}{c} 0\\ 1 \end{array}\right)\exp\left[iE_{0}\left(z-t\right)\right]$$

The 4-potential given by Equation (10) becomes:(19)$$\left(a_{0}^{\prime },a_{1}^{\prime },a_{2}^{\prime },a_{3}^{\prime }\right)=\left(0,-\frac{1}{f}\frac{\partial f}{\partial y},\frac{1}{f}\frac{\partial f}{\partial x},0\right)=-\left(a_{0},a_{1},a_{2},a_{3}\right)$$

Furthermore, according to Equation (13), the electromagnetic field corresponding to the above 4-potential takes the form:(20)$$E^{\prime }=0,B^{\prime }=\frac{1}{q}\frac{1}{f^{2}}\left[{\left(\frac{\partial f}{\partial x}\right)}^{2}+{\left(\frac{\partial f}{\partial y}\right)}^{2}-f\left(\frac{\partial^{2}f}{\partial x^{2}}+\frac{\partial^{2}f}{\partial y^{2}}\right)\right]k=-B$$

Thus, the electromagnetic 4-potential and field required to manipulate the transverse spatial distribution of Weyl particles with negative helicity is opposite to the one required to control the spatial distribution of particles with positive helicity.

Finally, a particularly important remark is that, according to Theorem 3.1 in [28], the spinors given by Equation (11) will also be solutions of the Weyl Equation (1) in [27] for an infinite number of 4-potentials, given by the formula: (21)$$b_{\mu }=a_{\mu }+\kappa_{\mu }s\left(r,t\right),\;\mu =0,1,2,3$$
where
(22)$$\left(\kappa_{0},\kappa_{1},\kappa_{2},\kappa_{3}\right)=\left(1,-\frac{\psi^{†}\sigma^{1}\psi }{\psi^{†}\psi },-\frac{\psi^{†}\sigma^{2}\psi }{\psi^{†}\psi },-\frac{\psi^{†}\sigma^{3}\psi }{\psi^{†}\psi }\right)=\left(1,0,0,-1\right)$$
and $s\left(r,t\right)$ is an arbitrary real function of the spatial coordinates and time. 

Similarly, in the case of particles with negative helicity, the spinors given by Equation (18) will also be solutions to the Weyl Equation (2) in [27] for the following 4-potentials:(23)$$b_{\mu }^{\prime }=a_{\mu }^{\prime }+\kappa_{\mu }^{\prime }s\left(r,t\right),\;\mu =0,1,2,3$$
where
(24)$$\left(\kappa_{0}^{\prime },\kappa_{1}^{\prime },\kappa_{2}^{\prime },\kappa_{3}^{\prime }\right)=\left(1,\frac{{\psi^{\prime }}^{†}\sigma^{1}\psi^{\prime }}{{\psi^{\prime }}^{†}\psi^{\prime }},\frac{{\psi^{\prime }}^{†}\sigma^{2}\psi^{\prime }}{{\psi^{\prime }}^{†}\psi^{\prime }},\frac{{\psi^{\prime }}^{†}\sigma^{3}\psi^{\prime }}{{\psi^{\prime }}^{†}\psi^{\prime }}\right)=\left(1,0,0,-1\right)=\left(\kappa_{0},\kappa_{1},\kappa_{2},\kappa_{3}\right)$$

According to Equation (13), the 4-potentials:(25)$$b_{\mu }-a_{\mu }=b_{\mu }^{\prime }-a_{\mu }^{\prime }=\left(1,0,0,-1\right)s\left(r,t\right)$$
correspond to the following electromagnetic fields:(26)$$\begin{array}{c} E_{s}\left(r,t\right)=-\frac{1}{q}\left(\frac{\partial s}{\partial x}i+\frac{\partial s}{\partial y}j+\left(\frac{\partial s}{\partial t}+\frac{\partial s}{\partial z}\right)k\right)\\ B_{s}\left(r,t\right)=\frac{1}{q}\left(\frac{\partial s}{\partial y}i-\frac{\partial s}{\partial x}j\right) \end{array}$$

This suggests that the state of Weyl particles will not be affected if any of the above electromagnetic fields are added to the magnetic fields given by Equations (14) and (20), corresponding to particles with positive and negative helicity, respectively. Thus, the process of controlling the spatial distribution of Weyl particles through appropriate magnetic fields is robust against electromagnetic perturbations, at least of the form described by Equation (26).

Finally, it should be noted that the proposed device does not have any special requirements regarding the transverse spatial distribution of Weyl particles. A uniform distribution would suffice; therefore, according to Equation (14), there is no need to apply any magnetic field, as the resulting magnetic field is zero for a constant function f. However, Equations (14) and (20) are expected to be very useful for other applications where the achievement of a specific transverse spatial distribution of Weyl particles is important.

## 4. Conclusions

In conclusion, we have described the principles of operation and the main properties of a simple and efficient device, the Weyl Parallel Switch (WPS), for controlling the flow of information based on Weyl fermions. This device has the advantage that it can control the flow of information along multiple channels simultaneously. In addition, the response time is exceptionally low, under 1 ps for typical values of the parameters, enabling the control of information flow at a rate of the order of 100 Petabits per second for a device with dimensions of the order of 1 cm, which is exceptional difficult to achieve using conventional electronics. Furthermore, the remarkable property of Weyl particles to be able to exist in the same quantum state under a wide variety of electromagnetic fields [27,28] provides enhanced robustness against electromagnetic perturbations, offering the opportunity to use the device in environments with high levels of electromagnetic noise. Consequently, the WPS is expected to play an important role in the emerging field of Weyltronics [4,5,6,7,8,9,10] and could be utilized in a variety of applications, such as telecommunications, signal processing, classical and quantum computing, etc. In addition, the WPS could also operate as a sensitive electric field sensor. Furthermore, we have proposed a method to fully control the transverse spatial distribution of Weyl fermions using appropriate magnetic fields, which could be used to guide Weyl fermions through the proposed device. Finally, in the Appendix A we discuss some very interesting remarks regarding the electromagnetic interactions of high-energy particles, where we have shown that the effects of degeneracy could also be applicable in this case.

## Figures and Tables

**Figure 1 sensors-24-03361-f001:**
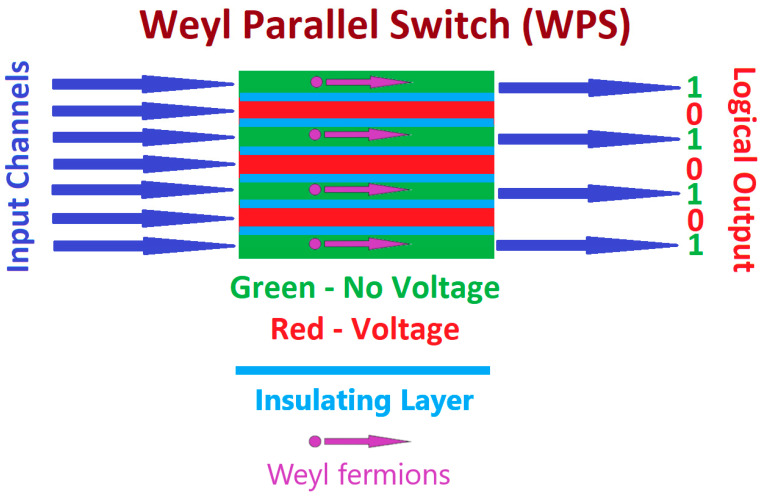
Schematic diagram of a device for controlling the flow of information based on Weyl particles.

## Data Availability

Data are contained within the article.

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
