# Peer review of "A Proposed Device for Controlling the Flow of Information Based on Weyl Fermions"

_sensors, 2024, doi:10.3390/s24113361_

Round 1

Reviewer 1 Report

Comments and Suggestions for Authors

This article proposes using appropriate electric and magnetic field to control the propagation of Weyl particles and presents a principle of operation of constructing and using such device (referred to as a Weyl Parallel Switch or WPS in the article) to control information flow simultaneously in multiple channels. This work is fully theoretical, which is within the aim and scope of Sensors. Despite the authors’ effort of providing equations, numerical examples, and other details concerning the proposed WPS, the reviewer feels that are still numerous points that remain unclear. Please see the following comments. There might be a potential for the proposed WPS to be practical in Weyltronics, as the authors anticipated and stated, the proposed device is still a little far from being realized in practice, given the fact that the article does not discuss any specific materials or more detailed designs as well as specifications. Despite this missing information, the reviewer thinks that there is value in this manuscript that may benefit broader audience. The reviewer suggests the authors to revise the manuscript by addressing the comments outlined below and, if possible, by also providing more specifics that could guide the realization of such device. In addition to the comments below, the reviewer has the impression that the number of references included in the current manuscript is small, which calls for the need of some additional literature survey of any relevant concept or demonstration.

1.       Though Ref. 11 has been published and can be used as the basis for applying an electric field perpendicular to the direction of motion of the Weyl fermion to control its propagation, it will be convenient for the reader that in the current manuscript the authors briefly discuss the reason behind choosing the electric field to be perpendicular than parallel, along with Eqs. (1) and (2).

2.       It will be helpful if simple schematic illustration of capacitors can be provided.

3.       It is not clear to the reviewer that, in the proposed WPS drawing in Figure 1, where the Weyl fermion propagates.

4.       The reviewer is curious if the Weyl particles propagate in different channels in Figure 1 can affect each other, which seems necessary to avoid.

5.       Though the authors stated that any materials that support the Weyl particle may be suitable for the proposed device and specifically emphasized the thin film, the reviewer was wondering how thick such thin films need to be?

6.       Are there any specific conditions needed to be satisfied for the proposed WPS to function, such as vacuum, low temperature, etc?

7.       Despite stating that the WPS consumes much low energy, is it possible to estimate and comment on the power consumption of the proposed WPS for controlling 100 Pbps information flow with 105 channels?

8.       How can the spatial-dependent magnetic field (seemingly on the x, y coordinates) be achieved especially when the footprint of the device is small?

9.       Could the authors comment on any potential limitations of the proposed WPS?

10.   The reviewer had a hard time finding a connection between section 4 “On the electromagnetic interaction of high energy particles” and the rest of the article, not to mention that this section does not discuss anything related to Weyl particles which the reviewer thinks of as the main point of this paper.

Reviewer 2 Report

Comments and Suggestions for Authors

Summary:

The authors build upon an earlier theoretical work to propose an electrical switching based on the electric field affected trajectory change of Weyl fermions. Because the Weyl fermions are massless, the authors show that in principle <1ps timescale switching can be achieved. Because of high degree of influence of the transverse electric field, the Weyl fermions can also constitute a sensing system with very high sensitivity. The control of the helicity with a transverse magnetic field is also discussed.

Overall, I think the manuscript proposes an interesting new idea. However, the physical argument to the operation of these devices, in my opinion, needs to be reinforced. I have the following remarks and suggestions.

Comment 1. I think the title of the manuscript saying “a novel device” is somewhat misleading. The authors propose only an idea of switching with Weyl fermions that should no way be taken as a proposed device. I believe some of the realistic considerations are- how the channel width affects the band-crossing and whether in a fabricated device that degeneracy would be broken. Also, since the Weyl particles are quasi-particle, what is the implication of their lifetime to the switching function, and at what temperature? In the drawing in Fig. 1 the channels are shown to touch each other laterally. Would that be the case for a real device?

Comment 2. The authors estimate the channel width by the minimum radius of curvature after application time of the electric field. However, the particles start at infinite radius of curvature trajectory and gradually spiral down to the lowest radius. Thus, for a channel width equal to the smallest radius, should the particles not already collide with the boundary at the start of the application of the E field?

Comment 3. Comment 2 also brings in the question—what is the physics of the Weyl particles colliding with the side walls of the channel? Is it a simple scattering process, or the particles are destroyed or decohered? Some discussion on this aspect will be very nice.

Comment 4. Section 4 of the manuscript does not really seem linked with the rest of the manuscript, except for the connection that Weyl fermions are massless, and it is not clear why it is included in the main body of the paper. This part, although interesting, is more like a textbook discussion. I think at best it could be put as an appendix/supplementary and not the main body. In the same way, I do not think “… some interesting remarks regarding the electromagnetic interactions…” does not constitute a scientific journal title.

Round 2

Reviewer 1 Report

Comments and Suggestions for Authors

The reviewer appreciates the response from the authors. The revised manuscript has addressed the reviewer's comments and thus the paper can be accepted.

Reviewer 2 Report

Comments and Suggestions for Authors

The authors have addressed all my comments. I have no further comments. I think the manuscript can be accepted.